# Malnutrition and its determinants among older adults living in foster families in Guadeloupe (French West Indies). A cross-sectional study

Denis Boucaud-Maitre[1,2,3]*, Nadine Simo-Tabue[4], Ludwig Mounsamy[3], Christine Rambhojan[3], Laurys Letchimy[4], Leila Rinaldo[3], Jean-François Dartigues[5], Hélène Amieva[5], Moustapha Dramé[2,4], Maturin Tabué-Teguo[2,4]

1 Centre Hospitalier Le Vinatier, Bron, France, 2 Equipe EPICLIV, Université des Antilles, Fort-de-France, Martinique, 3 Centre Hospitalo-Universitaire de Guadeloupe, Pointe-à-Pitre, Guadeloupe, 4 Centre Hospitalo-Universitaire de Martinique, Fort-de-France, Martinique, 5 Inserm U1219 Bordeaux Population Health Center, University of Bordeaux, Bordeaux, France

* denis.boucaud@gmail.com

**Data Availability Statement:** All relevant data are within the manuscript and its Supporting Information files.

## Abstract

### Background

Foster families may represent an alternative model for dependent older adults in many countries where nursing homes are insufficiently developed. This study aimed to assess the prevalence of malnutrition and its determinants in older adults living in foster families in Guadeloupe (French West Indies).

### Methods

This cross-sectional study was gathered from the KASAF (Karukera Study of Ageing in Foster families) study (n = 107, 41M/66F, Mdn 81.8 years). Nutritional status was assessed with the Mini Nutritional Assessment Short-Form (MNA-SF). Clinical characteristics and scores on geriatric scales (Mini-Mental State Examination (MMSE), Activities of Daily Living (ADL), Short Physical Performance Battery (SPPB), Center for Epidemiologic Studies- Depression (CESD) and Questionnaire Quality of Life Alzheimer's Disease (QoL-AD)) were extracted. Bivariate analysis and logistic models adjusted for age and gender were performed to test the association of nutritional status with socio-demographic variables and geriatric scales.

### Results

Thirty (28.0%) older adults were malnourished (MNA-SF score ≤7). In bivariate analysis, malnutrition was associated with an increased prevalence of cardiovascular diseases (46.7% versus 19.5%, p = 0.004), the presence of hemiplegia (30.0% versus 6.5%, p = 0.003), a poorer cognitive status (MMSE score 4.7 ± 7.1versus 9.7 ± 10.7; p = 0.031), higher risk of depression (CESD score 27.3 ± 23.0 versus 13.5 ± 14.4; p = 0.035) and dependency (ADL score 1.9 ± 1.9 versus 2.3 ± 2.1; p<0.001). Malnutrition was also associated with lower caregivers'rating of QoL (QoL-AD score 21.8 ± 6.4 versus 26.0 ± 5.7; p = 0.001) but not by

**Funding:** This study was supported by a grant from the Conseil Départemental de la Guadeloupe, ARS de la Guadeloupe, Saint-Martin, and Saint-Barthélemy (grant 2020/DPAPH/DRM). The funding body had no role in the design of the study and collection, analysis, and interpretation of data and in writing the manuscript.

**Competing interests:** The authors have declared that no competing interests exist.

older adult's rating (24.1 ± 11.2 versus 28.3 ± 7.7; p = 0.156). Similar associations were observed in logistic models adjusted for age and gender.

## Conclusion

Malnutrition was common among foster families for older adults. Special attention towards the prevention and treatment of malnutrition in older adults from cardiovascular diseases, cognitive impairment, dependency and depression is necessary in this model of dependency support.

## Introduction

Although foster families exist in many countries, this model is not widespread [1]. Consequently, very few studies have described and assessed the effectiveness of this model on potential adverse health outcomes [2]. In Guadeloupe (French West Indies, Caribbean Island), the number of foster families has increased over the past three decades, due to the aging of the population and the limited availability of nursing home placements. Cultural factors may also contribute to this phenomenon. For instance, the importance of the family in Caribbean culture and the public's reticence towards nursing homes may contribute to this phenomenon. Foster families assume responsibility for the care of one to three residents in their home, while a nurse, who visits the older adults on a daily basis, assumes paramedical care. Foster families are remunerated directly by the relevant public authorities. In a prospective observational study (KArukera Study of Ageing in Foster Families, KASAF), we observed that the profile of older adults in foster families was similar to that for older adults living in nursing homes in terms of co-morbidities, dementia and dependence [3]. Foster caregivers are responsible for the daily activities, including shopping, food preparation and the provision of meals. It is essential that these meals meet the nutritional needs of older adults.

Indeed, ensuring nutritional needs is fundamental to the care of older adults, particularly those who are dependent [4]. Ageing is associated with a change in body composition, a decrease in lean body mass and an increase in fat mass. Malnutrition predisposes older adults to an increased risk of adverse health outcomes such as frailty, osteoporosis, muscle wastage, mortality [4], a lack of energy [5], a decline in health and physical functions [6] or falls [7]. Malnultrition is frequently underestimated and neglected, as its manifestations are non-specific, particularly in the early stages. The following factors have been identified as increasing the risk of malnutrition: age over 85, low nutrient intake due to a loss of the ability to eat independently, difficulty swallowing, becoming bedridden, pressure ulcers, a history of hip fracture, dementia, depressive symptoms, and suffering from two or more chronic illnesses [8].

Malnutrition has been the subject of investigation in both nursing homes and the community. However, it has never been the subject of study in the context of foster families for older adults. The aim of this study was to estimate the prevalence of malnutrition among older adults receiving caregiving in foster families and to investigate the factors associated with malnutrition using the baseline data of the KASAF cohort.

## Methods

### Study design

KASAF cohort is a prospective observational study of older adults ($\geq$ 60 years old) living in foster families in Guadeloupe. The study protocol [9] and inclusion data has been published

[3]. At inclusion, 6 months and 12 months, healthcare professionals (geriatricians or clinical research nurses) interviewed the participants and their professional caregivers. For this study, we performed a cross-sectional analysis of the baseline's characteristics of participants. The KASAF study and was approved by the Sud Méditerranée III Ethics Committee on July 1, 2020 (number 2020.05.03 bis_ 20.04.01.59610).

## Outcome measure

The nutritional status was evaluated using the Mini Nutritional Assessment Short-Form (MNA-SF) (Rubenstein) [10]. 15 The MNA-SF comprises six items: reduced food intake, non-volitional weight loss in the past 3 months, mobility, psychological stress or acute disease during the past 3 months, neuropsychological problems, and low body mass index (BMI). For adults whose BMI was missing, it was replaced by low calf circumference, as recommended in the MNA-SF guidance [10]. The total MNA-SF score ranges from 0 (indicating the most severe form of malnutrition) to 14 (indicating no sign of malnutrition). In particular, a score of 12–14 is indicative of a normal nutritional status, while a score of 0–7 and 8–11 identifies malnutrition or risk of malnutrition respectively.

## Other measurements

The sociodemographic data and comorbidities were collected from the foster caregiver. The cognitive status was assessed using the Mini-Mental State Examination (MMSE) [11]. A score below 18 indicated the presence of major cognitive impairment. Functional status was evaluated using the Activities of Daily Living (ADL) scale [12] and the instrumental ADL scale (IADL) [13]. Physical function was assessed using the Short Physical Performance Battery (SPPB) [14] and depression with the Center for Epidemiologic Studies Depression (CESD) scale [15]. Quality of life of the participant was assessed using the QoL-AD (Questionnaire Quality of Life—Alzheimer's Disease) [16], which was administered to the participant and the caregiver. Pain was quantified using a visual analogue scale (VAS), with scores ranging from 0 to 100.

## Statistical analysis

Quantitative variables were expressed as mean ± standard deviation, median and minimum and–maximum values. The qualitative variables were expressed as percentages. Chi-square or Fisher test and t-tests were used to describe the population according to their nutritional status. A Pearson correlation test was used to assess the correlation between the QOL-AD scores of patient and their respective caregiver. Logistic regression models, which were adjusted for age and gender, were conducted to examine the association between nutritional status (the independent variable) and each comorbidity and each geriatric scale. We reported odds ratios (ORs) and 95% confidence intervals (95% CIs). No imputation method was performed for missing data. Statistical significance was set at $P < 0.05$. All analyses were performed with R. 4.2.1.

## Results

### 1. Frequency of malnutrition

A total of 107 older adults were included in the study. The mean age was 82.2 ± 11.6 years, and 38.3% of the participants were men. They had been living in foster care for 4.6 ± 4.8 years. The frequency of malnutrition (MNA-SF ≤ 7) was 28.0% (95% confidence interval (CI): 20.9–39.1) (n = 30). Furthermore, 52 (48.6%, IC95%: 39.1–58.1) older adults were at risk of malnutrition

(MNA-SF between 8 and 11 points). The prevalence of older adults with malnutrition or at risk of malnutrition was 76.6% (IC95%: 68.6–84.6).

## 2. Factors associated with sociodemographic status and comorbidities

In bivariate analysis, malnutrition (compared to normal nutritional status or at risk of malnutrition) was associated with cardiovascular diseases (46.7% versus 19.5%, p = 0.004) and hemiplegia (30.0% versus 6.5%, p = 0.003). Malnutrition was not associated with age, gender, length of stay in foster families, hypertension, diabetes, dementia and Parkinson's disease listed by the caregiver (Table 1). In a model adjusted for age and gender, the OR were 3.94 (CI95%: 1.52–10.62) for cardiovascular disease and 11.36 (CI95%: 3.00–53.29) for hemiplegia.

## 3. Association between MNA-SF score and geriatric scales

Malnutrition (compared to normal nutritional status or at risk of malnutrition) was associated with poorer cognitive status assessed by the MMSE score (4.7 ± 7.1 versus 9.7 ± 10.7, p = 0.031), especially among older adults with major cognitive disorders (MMSE score < 18) (92.3% versus 30.0%, p = 0.023). Among the 28 older adults with a MMSE score <18 who were not diagnosed with dementia by the caregiver, six were malnourished (21.4%). Malnutrition was also associated with a lower ADL score (1.9 ± 1.9 versus 2.3 ± 2.1, p<0.001). Malnutrition was highly associated with bedridden older adults (96.7% versus 67.5%, p = 0.001) and older adults totally dependent at meals (80.0% versus 42.9%, p<0.001) in terms of activities of daily living. Malnutrition was associated with the caregivers 'estimation of QOL score (QoL-AD score 21.8 ± 6.4 versus 26.0 ± 5.7; p = 0.001) but not by the self-reported QoL score

**Table 1. Sociodemographic factors and comorbidities associated with nutritional status in KASAF study.**

| Characteristics | All (n = 107) | Malnutrition yes (n = 30) | Malnutrition no (n = 77) | p | OR (CI95%) | p |
|---|---|---|---|---|---|---|
| | | Bivariate analysis | | | Model adjusted on age and gender | |
| Age | 82.2 ± 11.6 | 84.7 ± 11.3 | 81.2 ± 11.2 | 0.156 | | |
| <80 years old | 44 (41.1%) | 8 (18.2%) | 36 (81.8%) | | | |
| ≥ 80 years | 63 (58.9%) | 22 (34.9%) | 41 (65.1%) | 0.057 | | |
| Gender (men) | 41 (38.3%) | 8 (26.7%) | 33 (42.9%) | 0.122 | | |
| Length of stay in foster families | 4.6 ± 4.8 | 4.2 ± 3.7 | 4.8 ± 5.2 | 0.576 | 0.98 (0.88–1.07) | 0.647 |
| Hypertension | 49 (45.8%) | 14 (46.7%) | 35 (45.4%) | 0.910 | 1.02 (0.42–2.42) | 0.959 |
| Diabetes | 26 (24.3%) | 7 (23.3%) | 19 (24.7%) | 0.884 | 0.93 (0.32–2.51) | 0.903 |
| Hypercholesterolemia | 11 (10.3%) | 4 (13.3%) | 7 (9.1%) | 0.498 | 2.86 (0.59–13.63) | 0.178 |
| Cardiovascular diseases (cardiac failure, myocardial infarction, stroke) | 29 (27.1%) | 14 (46.7%) | 15 (19.5%) | 0.004 | 3.94 (1.52–10.62) | 0.005[i] |
| Dementia | 53 (49.5%) | 19 (63.3%) | 34 (44.2%) | 0.075 | 1.75 (0.70–4.52) | 0.237 |
| Parkinson's disease | 13 (12.2%) | 6 (20.0%) | 7 (9.1%) | 0.184 | 2.58 (0.75–8.80) | 0.124 |
| Hemiplegia | 14 (13.1%) | 9 (30.0%) | 5 (6.5%) | 0.003 | 11.36 (3.00–53.29) | <0.001[ii] |
| Kidney disease | 4 (3.7%) | 3 (10.0%) | 1 (1.3%) | 0.066 | 6.84 (0.81 (144.09) | 0.107 |
| Cancer | 1 (0.9%) | 0 (0.0%) | 1 (1.3%) | - | | |

[i]: McFadden's Pseudo R2: 0.090

[ii]: McFadden's Pseudo R2: 0.133

**Table 2. Associations between MNA-SF score and geriatric scales.**

| Scale | All (n = 107) | Bivariate analysis | | | Model adjusted on age and gender | |
| | | Malnutrition yes (n = 30) | Malnutrition no (n = 77) | p | OR | p |
| --- | --- | --- | --- | --- | --- | --- |
| MMSE (n = 96) | 8.3 ± 10.1 | 4.7 ± 7.1 | 9.7 ± 10.7 | 0.031 | 0.94 (0.89–0.99) | 0.045[1] |
| MMSE≤18 (n = 96) | 73 (76.0%) | 24 (92.3%) | 21 (30.0%) | 0.023 | 4.92 (1.27–32.69) | 0.043 |
| ADL (n = 107) | 1.5 ± 1.8 | 1.9 ± 1.9 | 2.3 ± 2.1 | <0.001 | 0.51 (0.31–0.76) | 0.004[2] |
| *Full assistance for bathing* | 84 (78.5%) | 28 (93.3%) | 56 (72.7%) | 0.020 | | |
| *Full assistance of dressing* | 88 (82.2%) | 29 (96.7%) | 59 (76.6%) | 0.015 | | |
| *Full assistance for toileting* | 90 (84.1%) | 29 (96.7%) | 61 (79.2%) | 0.015 | | |
| *Bedridden* | 81 (75.7%) | 29 (96.7%) | 52 (67.5%) | 0.001 | | |
| *Incontinence* | 91 (85.0%) | 29 (96.7%) | 62 (80.5%) | 0.035 | | |
| *Totally dependent at meals* | 57 (53.3%) | 24 (80.0%) | 33 (42.9%) | <0.001 | | |
| QOL-AD (n = 47) residents | 27.2 ± 8.8 | 24.1 ± 11.2 | 28.3 ± 7.7 | 0.156 | 0.93 (0.83–1.01) | 0.104 |
| QOL-AD caregivers'estimation (n = 47) | 24.8 ± 6.2 | 21.8 ± 6.4 | 26.0 ± 5.7 | 0.001 | 0.87 (0.79–0.95) | 0.003[3] |
| VAS pain (n = 37) | 44.9 ± 35.6 | 63.3 ± 2.6 | 41.3 ± 36.4 | 0.169 | 1.01 (0.99–1.04) | 0.322 |
| SPPB (n = 105) | 1.0 ± 2.0 | 0.4 ± 1.3 | 1.2 ± 2.2 | 0.07 | 0.78 (0.52–1.03) | 0.147 |
| CESD (n = 39) | 16.7 ± 17.4 | 27.3 ± 23.0 | 13.5 ± 14.4 | 0.035 | 1.05 (1.00–1.11) | 0.031[4] |

[1]: McFadden's Pseudo R2: 0.178

[2]: McFadden's Pseudo R2: 0.133

[3]: McFadden's Pseudo R2: 0.110

[4]. McFadden's Pseudo R2: 0.724

(24.1 ± 11.2 versus 28.3 ± 7.7; p = 0.156). The correlation coefficient between the QoL-AD score for older adult and their respective caregivers was 0.60 (p<0.001). Finally, the CESD score for depression was associated with malnutrition (27.3 ± 23.0 versus 13.5 ± 14.4; p = 0.035) (Table 2). The SPPB score (0.4 ± 1.3 versus 1.2 ± 2.2; p = 0.07) and VAS pain score (63.3 ± 2.6 versus 41.3 ± 36.4, p = 0.169) were not statistically associated with the MNA-SF score (Table 2).

In model adjusted for age and gender, the OR for malnutrition was 4.92 (1.27–32.69) for a MMSE score of ≤18, 0.51 (0.31–0.76) for the ADL score, 0.87 (0.79–0.95) for QOL-AD caregivers 'estimation and 1.05 (1.00–1.11) for the CESD score.

## Discussion

This is the first study to assess malnutrition in foster families for dependent older adults. The results highlighted the high prevalence of malnutrition in this setting (28.0%). In community-dwelling older adults, the prevalence of malnutrition is between 3 to 6%, depending on the setting and assessment method [4, 17, 21]. In Guadeloupe, the prevalence of malnutrition or at-risk of malnutrition in older adults is 21.7% at home [18], which is a significantly lower than observed in our study (i.e. 76.6%). Foster families in Guadeloupe are considered an alternative to nursing homes. In the literature, the frequency of malnutrition in nursing homes, based on the MNA scale, is estimated at 13.8% [17]. In France, a study carried out in nursing homes found a frequency of 15.7% [19]. We observed a frequency of malnutrition of 92.3% in older adults with severe cognitive impairment, which appears to be higher than that reported in the literature. The estimated range is 6.8% to 75.6% [20] or 28.7% in another systematic review using only the MNA score [21]. In the model adjusted for age and gender, a MMSE score ≤18 was associated with malnutrition (OR: 4.92 (CI95%: 1.27–32.69)). The finding of the study

indicated that dementia, as reported by the foster caregiver, was not associated with malnutrition. However, the MMSE score suggested that almost 20% of the older adults suffered from undetected severe cognitive impairment. Dementia, as well as undernutrition, seems to be underestimated by foster caregivers.

Malnutrition was particularly prevalent in older adults with a history of cardiovascular disease and hemiplegia. It is well established that malnutrition increases the risk of mortality and hospitalizations in patients with chronic heart failure [22]. Our study is consistent with several other studies conducted in nursing homes that have investigated the potential association between malnutrition and depression or poor physical function [23]. Furthermore, we observed a strong association between dependency and malnutrition, particularly for in patients who are bedridden or have difficulty eating. With regard to quality of life, we noted that malnutrition was associated with QoL score as perceived by caregivers, but not with that rated by older adults themselves. In nursing homes, malnutrition impacts quality of life [24, 25]. This result may be due to the low number of older adults who were able to answer to the QoL-AD scale, excluding older adults with severe dementia. Impaired cognition has been associated with reduced quality of life when the caregiver is the assessor [26]. Moreover, quality of life perceived by the older adult is generally rated higher than that perceived by the proxies' rating [27, 28]. Higher prevalence of malnutrition have been observed in adults aged > 80 years and women [29]. Although the association was not statistically significant, the frequency of malnutrition was higher in adults aged > 80 years (34.9% versus 18.2%, p = 0.057) in our study.

Our study therefore provides important elements for the assessment of the foster family model for dependent older adults. One strength of our study is that it presents data from a population of Caribbean population, with a specific diet (especially in terms of fruit and vegetables) and probably specific dietary intake [30]. Foster families for older adults could provide a solution to the challenge of dependency in many countries, particularly in the Caribbean and Africa. Improving nutritional care represents an essential lever for developing this model. In terms of nutrition, the foster family presents both a strength and a weakness. It is easier to respect the food tastes and preferences of the older adult in a domestic setting than in a collective kitchen such as those found in nursing homes. Furthermore, the residents of nursing homes have less flexibility in their meal schedules. Nevertheless, the quantity and quality of home-cooked meals may not be optimal for malnourished older adults. Additional training and specialized dietetic care, including advice, food enrichment, anthropometric monitoring, consultations with nutritionists and dieticians and a food diary [4], could be provided if malnutrition is detected. An alternative solution could be the implementation of meal delivery services. Currently, in Guadeloupe, the authorization to work as a foster caregiver requires 54 hours of training, with only a few hours devoted to hygiene and nutrition. It is also noteworthy that weight was only available for 22 participants, despite the simplicity of the tool for detecting recent malnutrition. Paramedical staff could also provide training and screening for malnutrition, given that all foster care residents benefit from a daily visit from a nurse.

Our study has a number of limitations. Firstly, there were no data concerning the precariousness of family caregivers and the budget allocated to buying meals for the older people. This socio-economic data could have been interesting to explore. Secondly, due to the low sample size and the limit number of outcomes events for CESD scale or QOL-AD scale, no multivariate model taking into account all the covariates associated with malnutrition was performed. This is a cross-sectional study suggesting associations. The one-year longitudinal follow-up of our study will enable us to identify risk factors for nutritional deterioration, including hospitalizations and ADL.

## Conclusion

Malnutrition was common among older adults living in foster families. The prevalence of malnutrition was higher in older adults with dependency, depression, cardiovascular diseases, hemiplegia and cognitive impairment. The findings of this study indicate that there is a need for greater focus on the nutritional requirements of older adults and the training of foster caregivers in this area.

## Supporting information

**S1 Checklist. Human participants research checklist.**
(DOCX)

**S1 Dataset.**
(CSV)

## Author Contributions

**Conceptualization:** Denis Boucaud-Maitre, Jean-François Dartigues, Hélène Amieva, Maturin Tabué-Teguo.

**Data curation:** Denis Boucaud-Maitre.

**Formal analysis:** Denis Boucaud-Maitre, Christine Rambhojan.

**Funding acquisition:** Maturin Tabué-Teguo.

**Investigation:** Denis Boucaud-Maitre, Nadine Simo-Tabue, Maturin Tabué-Teguo.

**Methodology:** Denis Boucaud-Maitre, Jean-François Dartigues, Hélène Amieva, Maturin Tabué-Teguo.

**Project administration:** Christine Rambhojan.

**Supervision:** Nadine Simo-Tabue, Jean-François Dartigues, Hélène Amieva, Moustapha Dramé, Maturin Tabué-Teguo.

**Validation:** Jean-François Dartigues, Hélène Amieva, Moustapha Dramé, Maturin Tabué-Teguo.

**Writing – original draft:** Denis Boucaud-Maitre, Nadine Simo-Tabue.

**Writing – review & editing:** Ludwig Mounsamy, Christine Rambhojan, Laurys Letchimy, Leila Rinaldo, Jean-François Dartigues, Hélène Amieva, Moustapha Dramé, Maturin Tabué-Teguo.

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
