## [Decision Letter · Decision Letter 0]

18 Mar 2024

PONE-D-24-00892Malnutrition and its determinants among older adults living in foster families in Guadeloupe (French West Indies): A cross-sectional study.PLOS ONE

Dear Dr. Boucaud-Maitre,

Thank you for submitting your manuscript to PLOS ONE. After careful consideration, we feel that it has merit but does not fully meet PLOS ONE’s publication criteria as it currently stands. Therefore, we invite you to submit a revised version of the manuscript that addresses the points raised during the review process.  Though  the  manuscript has scientific merit , it  requires major revision before being considered for publication . Therefore , please submit your revised manuscript by May 02 2024 11:59PM. If you will need more time than this to complete your revisions, please reply to this message or contact the journal office at plosone@plos.org. Please include the following items when submitting your revised manuscript:A rebuttal letter that responds to each point raised by the academic editor and reviewer(s). You should upload this letter as a separate file labeled 'Response to Reviewers'.A marked-up copy of your manuscript that highlights changes made to the original version. You should upload this as a separate file labeled 'Revised Manuscript with Track Changes'.An unmarked version of your revised paper without tracked changes. You should upload this as a separate file labeled 'Manuscript'.

We look forward to receiving your revised manuscript.

Kind regards,

Gudina Egata, PhD in Public Health

Academic Editor

PLOS ONE

**Journal requirements: **

2.  In the online submission form you indicate that your data is not available for proprietary reasons and have provided a contact point for accessing this data. Please note that your current contact point is a co-author on this manuscript. According to our Data Policy, the contact point must not be an author on the manuscript and must be an institutional contact, ideally not an individual. Please revise your data statement to a non-author institutional point of contact, such as a data access or ethics committee, and send this to us via return email. Please also include contact information for the third party organization, and please include the full citation of where the data can be found.

Reviewers' comments:

Reviewer's Responses to Questions

**Comments to the Author**

1. Is the manuscript technically sound, and do the data support the conclusions?

Reviewer #1: Partly

2. Has the statistical analysis been performed appropriately and rigorously? 

Reviewer #1: Yes

3. Have the authors made all data underlying the findings in their manuscript fully available?

Reviewer #1: Yes

4. Is the manuscript presented in an intelligible fashion and written in standard English?

Reviewer #1: Yes

5. Review Comments to the Author

Reviewer #1: Firstly, I want to thank the authors for bringing out the issue of nutrition and health among older adults residing in foster care homes. Studies on institutionalized and community-based nutritional survey among older adults have been many, however the current study is a unique in that it presents data on older adults living in foster families, which is an important issue. In below points, I have advised important comments and constructive feedback on the manuscript.

Abstract:

please modify and rewrite conclusion statements as they are not what the findings show.

Introduction:

Line num 62- I think the authors meant “prevalence of malnutrition among older adults receiving caregiving in foster families”.

Methods:

Line num 72- please correct the grammatical mistake in the sentence.

Line num 76- "15"...?

Statistical analysis:

Line num 102- please mention in full which version of R studio and all R packages that were used.

Results:

Please report the prevalence data on all three groups.

Did you check for the confounders? If yes, which procedure did you use to do it? Please explain this in the method section. Which variables were used for multivariate analysis and what was the criteria for this? Why did the authors adjust for age and gender?

Please also mention the adjusted R2 values for each of the variables that had statistically significant association with the dependent variable.

Discussion:

Line num 146- please use the term "prevalence" instead of "frequency".

Line num 147- please add more references.

Conclusion:

As mentioned in the above comment, please modify the conclusion statements. The conclusion has not been specifically written based on the findings. The findings show association between predictors(CVD events, hemiplegia) and nutritional status and the authors have not presented any data on the assessment of knowledge, attitude, training, education of the caregivers. This will limit the authors from saying that "training of caregivers are needed". Had the authors presented data on caregiver's level of knowledge or training, this could have led authors recommend training of caregivers although training might be an important factor that impacts older adults' health and life.

6. PLOS authors have the option to publish the peer review history of their article (what does this mean?). If published, this will include your full peer review and any attached files.

Reviewer #1: **Yes: **Man Kumar Tamang

---

## [Author Response · Author response to Decision Letter 0]

3 May 2024

Reviewer #1: Firstly, I want to thank the authors for bringing out the issue of nutrition and health among older adults residing in foster care homes. Studies on institutionalized and community-based nutritional survey among older adults have been many, however the current study is a unique in that it presents data on older adults living in foster families, which is an important issue. In below points, I have advised important comments and constructive feedback on the manuscript.

Authors comment: We thank the reviewer for her/his encouraging comments and its interest in this model of accommodation/care for older adults. 

Abstract:

1. Please modify and rewrite conclusion statements as they are not what the findings show.

Authors Response: We agree with the reviewer’s comment. We propose the following conclusion: “Special attention towards the prevention and treatment of malnutrition in older adults from cardiovascular diseases, cognitive impairment, dependency and depression is necessary in this model of dependency support.”

Introduction:

2. Line num 62- I think the authors meant “prevalence of malnutrition among older adults receiving caregiving in foster families”.

Authors Response: We agree with the reviewer’s comment. We have changed the sentence as proposed by the reviewer.

Methods:

3.Line num 72- please correct the grammatical mistake in the sentence.

Authors Response: We rewrote the sentence as requested.

4.Line num 76- "15"...?

Authors Response: The number authorization is well 2020.05.03 bis_ 20.04.01.59610

Statistical analysis:

5.Line num 102- please mention in full which version of R studio and all R packages that were used.

Authors Response: The full version of R was R. 4.2.1 (added to the manuscript). No specific package was used.

Results:

6.Please report the prevalence data on all three groups.

Authors Response: Agree, see manuscript.

7. Did you check for the confounders? If yes, which procedure did you use to do it? Please explain this in the method section. Which variables were used for multivariate analysis and what was the criteria for this? Why did the authors adjust for age and gender?

Authors Response: We did not perform multivariate analysis with all the variables associated with malnutrition in bivariate analysis (p<0.2) or with a stepwise/backward selection for example. Indeed, we consider that the number of participants was too low to perform multivariate analysis with several covariables. Half of participants (i.e. older adults with major cognitive impairment) were unable to answer to CESD scale or quality of life scale. Consequently, a multivariate analysis including for example MMSE score, ADL score, age, sex, cardiovascular disease, hemiplegia, SPPB and CESD (variables associated with malnutrition in bivariate analysis with p<0.2) would have include less than half the participants (n=37). In statistics, to carry out a linear or logistic regression, it is recommended to have at least 10 observations per covariate (Peduzzi, P., Concato, J., Kemper, E., Holford, T. R. & Feinstein, A. R. A simulation study of the number of events per variable in logistic regression analysis. J Clin Epidemiol 49, 1373–1379 (1996). We could perform multivariate analysis without CESD, but depression being strongly associated with malnutrition, we consider that this will be not relevant to exclude depression in a multivariate analysis. It is why we decided to perform only logistic regression models adjusted on age and sex for each variable. We have better explained the methodology section and added this point as a limit of our study (see manuscript).

8.Please also mention the adjusted R2 values for each of the variables that had statistically significant association with the dependent variable.

Authors Response: We have mentioned McFadden's Pseudo R2 as requested.

Discussion:

9. Line num 146- please use the term "prevalence" instead of "frequency".

Authors Response: Agree

10. Line num 147- please add more references.

Authors Response: Agree, we have added cross-reference to a recent review (2023) published by the Lancet: Dent E, Wright ORL, Woo J, Hoogendijk EO. Malnutrition in older adults. Lancet. 2023 Mar 18;401(10380):951-966 

and another systematic review of malnutrition using the MNA scale: Cereda E, Pedrolli C, Klersy C, Bonardi C, Quarleri L, Cappello S, Turri A, Rondanelli M, Caccialanza R. Nutritional status in older persons according to healthcare setting: A systematic review and meta-analysis of prevalence data using MNA®. Clin Nutr. 2016 Dec;35(6):1282-1290.

Conclusion:

11. As mentioned in the above comment, please modify the conclusion statements. The conclusion has not been specifically written based on the findings. The findings show association between predictors(CVD events, hemiplegia) and nutritional status and the authors have not presented any data on the assessment of knowledge, attitude, training, education of the caregivers. This will limit the authors from saying that "training of caregivers are needed". Had the authors presented data on caregiver's level of knowledge or training, this could have led authors recommend training of caregivers although training might be an important factor that impacts older adults' health and life.

Authors Response: We fully agree with the reviewer. Our conclusion should more reflected the results of our study and not our perspective of work! Indeed, we project to perform a one-year randomized clinical trial in foster families to assess the efficacy of a training of caregivers by a nutritionist compared to no intervention. We have changed the conclusion as requested.

---

## [Editor Report · Decision Letter 1]

22 May 2024

Malnutrition and its determinants among older adults living in foster families in Guadeloupe (French West Indies): A cross-sectional study.

PONE-D-24-00892R1

Dear  Denis Boucaud-Maitre,

We’re pleased to inform you that your manuscript has been judged scientifically suitable for publication and will be formally accepted for publication once it meets all outstanding technical requirements.

Kind regards,

Gudina Egata, PhD in Public Health

Academic Editor

PLOS ONE
---

## [Editor Report · Acceptance letter]

27 May 2024

PONE-D-24-00892R1 

PLOS ONE

Dear Dr. Boucaud-Maitre, 

I'm pleased to inform you that your manuscript has been deemed suitable for publication in PLOS ONE. Congratulations! Your manuscript is now being handed over to our production team.

Kind regards, 

on behalf of

Dr. Gudina Egata 

Academic Editor

PLOS ONE